# Low Dielectric Constant Polyimide Obtained by Four Kinds of Irradiation Sources

**DOI:** 10.3390/polym12040879

**Published:** 2020-04-10

**Authors:** Hongxia Li, Jianqun Yang, Shangli Dong, Feng Tian, Xingji Li

**Affiliations:** 1School of Material Science and Engineering, Harbin Institute of Technology, Harbin 150001, China; koalalhx@163.com (H.L.); yangjianqun@hit.edu.cn (J.Y.); sldong@hit.edu.cn (S.D.); 2Shanghai Synchrotron Radiation Facility, Zhangjiang Lab, Shanghai Advanced Research Institute, Chinese Academy of Sciences, Shanghai 201204, China; tianfeng@zjlab.org.cn

**Keywords:** low dielectric constant, PI, irradiation, dielectric loss

## Abstract

Irradiation is a good modification technique, which can be used to modify the electrical properties, mechanical properties, and thermal properties of polymer materials. The effects of irradiation on the electrical properties, mechanical properties, and structure of polyimide (PI) films were studied. PI films were irradiated by a 1 MeV electron, 3 MeV proton, 10 MeV proton, and 25 MeV carbon ion. Dielectric constant, dielectric loss, and resistance measurements were carried out to evaluate the changes in the electrical properties; moreover, the mechanical properties of the pristine and irradiated PI were analyzed by the tensile testing system. The irradiation induced chemical bonds and free radicals changes of the PI films were confirmed by the Fourier transform infrared (FTIR) spectra, X-ray photoelectron spectroscopy (XPS), and electron paramagnetic resonance (EPR). The dielectric constant of the PI films decreases with the increase of fluences by the four kinds of irradiation sources.

## 1. Introduction

In recent years, with the rapid development of microelectronics industry, the miniaturization of electronic components, the rapid growth of large scale integrated circuit chips, the increase of chip interconnect density, and the improvement the circuit connection of the resistance and capacitance, the signal delay of resistance production effect signal transmission speed and signal loss [1,2,3]. As a kind of functional material, low dielectric constant polymers have become an important research direction. Polyimide, as an important insulating and encapsulating material, is widely used in aerospace and microelectronic fields. Currently, many researchers have reported that polyimide (PI) thin films with low dielectric constant are obtained by doped fluorine-containing groups, doped porous, or other introduced functional groups methods. Goto et al. [4,5,6,7] successfully prepared a series of thermostable low dielectric constant polyimide (PI) by introducing diphenyl fluorine group into the main chain. The result shows that the lowest dielectric constant of the non-fluorinated PI is 2.77, while the lowest dielectric constant of the fluorinated PI is 2.35. Fujiwara et al. [8] prepared poly(norborneolin imide) with large fluorinated aromatic groups in the side chain and confirmed that the main chain structure of fluorinated poly(norborneolin imide) showed helical structure through molecular simulation. The results showed that the polymer had good heat resistance and glass transition temperature (*T*_g_) available at above 400 °C, the spiral structure increased the free volume, and the dielectric constant was as low as 2.31 [9]. The reason for the decrease in the dielectric constant of the PI-PDMS polymer is that polydimethylsiloxane (PDMS) was added to the polyimide, and then the chain reaction produced carbon dioxide to form nanoporous films. The result showed that the copolymers had the lowest dielectric constant (2.58) when the mass fraction ω(PDMS) = 25% and ω(MDI) = 5%. Meanwhile, Lee et al. [10,11,12] prepared nanoporous polyimide films with polyoxyethylene (PEO) polyhedral oligomeric silsesquioxane (POSS) nanoparticles, and the dielectric constant of such porous hybrid films were 3.25–2.25. The method can be used to obtain low dielectric constant polymers by doping porous materials or introducing other functional groups.

Radiation treatment of polymers involves irradiation of the polymers, usually in a continuous mode, and modification of the polymers to improve their performance for industrial purposes. The irradiation processing of polymers mainly includes cross-linking, curing, grafting, and degradation [13,14,15,16]. Irradiation changes the structure and properties of polymer materials, and the dielectric constant of the irradiated polymer decreases when the content of polarized group decreases. Polyimide (PI) can be used as dielectric material for thin-film transistors and capacitors [17,18,19,20]. Therefore, it is of great significance to study the properties of irradiated polymers to evaluate the improvement or deterioration in their mechanical strength, molecular weight, dielectric properties, and thermal properties. At present, some researchers have reported the effects of irradiation on the dielectric properties of polymers. S. Raghu et al. [21] used electron beam and gamma ray irradiated polymer electrolyte films, the results showed that the dielectric constant and conductivity increased with the increase of irradiation fluences. Qureshi et al. [22] found that the dielectric constant and dielectric loss of polyimide increased with the increase of irradiation fluences after irradiation with 80 MeV O^6+^ ion. Quamara et al. [23] pointed out that the dielectric behavior of polyimide was a non-monotonic evolution law with the increase of irradiation fluences after 50 MeV Si^+^ ion irradiation and that it was temperature-dependent. However, the irradiated polymers with low dielectric constant are rarely reported.

In this research work, PI films were exposed by irradiation sources including a 1 MeV electron, 3 MeV proton, 10 MeV proton, and 25 MeV carbon ion with different fluences. The dielectric properties of pristine and irradiated PI films were analyzed by dielectric spectroscopy, and it was found that the dielectric constant decreased to a minimum of 2.7. However, the mechanical properties did not change much compared with pristine PI. The four kinds of irradiation sources can decrease the dielectric constant of PI films.

## 2. Experimental Section

### 2.1. Materials and Equipment

Kapton-H polyimide (PI, Isophthalic anhydride diaminodiphenyl ether) films with the density of 1.34 g/cm^3^ and 50 μm in thickness were purchased from Du Pont co., Ltd., with molecular formula (C_22_H_10_N_2_O_5_)_n_. The high-energy electron irradiation test was performed by the DD 1.2 high-frequency high-voltage electronic accelerator in Heilongjiang Technical Physics Institute. High-energy proton and carbon ion irradiation experiments were carried out using the EN2 × 6 tandem electrostatic accelerator at Peking University.

### 2.2. Experimental Parameter

The PI films were cut into 5 × 5 cm^2^ size and all the irradiation area was larger than this size. The electron energy is 1 MeV; particle radiation flux is 2 × 10^8^ cm^−2^·s^−1^; and the irradiation fluences are 3.8 × 10^13^, 3.8 × 10^14^, and 2.2 × 10^16^ e/cm^2^, respectively. The 1 MeV electron penetration depth is ~100 μm calculated by Geant4. The proton energy is 3 MeV; the 3 MeV proton penetration depth is ~100 μm, as calculated by Stopping and Ranges of Ions in Matter (SRIM) software; and the irradiation fluences are 4.4 × 10^11^, 1.1 × 10^12^, and 2.2 × 10^12^ p/cm^2^, respectively. The proton energy is 10 MeV; the 10 MeV proton penetration depth is ~1000 μm, as calculated by SRIM software; and the irradiation fluences are 5 × 10^11^, 1 × 10^12^, and 1 × 10^13^ p/cm^2^, respectively. The carbon ion energy is 25 MeV, the 25 MeV carbon ion penetration depth is ~30 μm as calculated by SRIM software, and the films thickness is 50 μm, thus the 25 MeV carbon ion does not penetrate through the PI films, and the irradiation fluences are 2.8 × 10^11^, 1.2 × 10^12^, and 5 × 10^12^ ion/cm^2^, respectively. The sizes for the dumbbell-shaped tensile samples are 12, 4, and 0.5 mm.

### 2.3. Characterizations

The dielectric constant, dielectric loss, and insulation resistance of the samples were measured by BDS 4000 Novocontrol broadband dielectric spectrometer/impedance, which was produced in Germany in a wide frequency range from 10^−1^ to 10^7^ Hz at room temperature. Both sides of the PI samples were plated with a 25 mm diameter Al electrode by the vacuum evaporation machine before test. The equipment used in external stimulus is an electric field, E. The composition and chemical state of polyimide were characterized by the X-ray photoelectron spectroscopy (XPS) (K-Alpha) produced by the Thermal fisher scientific. The samples used for the tensile test are dumbbell-shaped samples with drawing rate of 2 μm/s at room temperature in air, which uses an MST810 material tensile testing system. Fourier infrared spectroscopy (FTIR) spectra of films have been obtained by Nicolet Magna-IR 560 spectrometer operated in transmission mode with the spectral region between 4000 and 700 cm^−1^ and scanned 32 times on average. Free radicals were tested by the A200 type Electron paramagnetic resonance (EPR) from the German Bruke company. In this experiment, the center of the magnetic field is 3580 G, the sweep width field is 90 G, the microwave frequency is 9.85 GHz, the time constant is 20 s, the spectral gain is 2 × 10^4^, the modulation amplitude is 1 G, and the microwave power is 6.140 mW.

## 3. Results

### 3.1. Dielectric Constant Analysis

The dielectric constant (ε) of pristine and irradiated PI samples after 1 MeV electron with different fluences (3.8 × 10^13^ e/cm^2^, 3.8 × 10^14^ e/cm^2^, 2.2 × 10^16^ e/cm^2^) at different frequency and room temperature are shown in Figure 1a. The dielectric constant of PI films decreases after the four kinds of irradiation sources; moreover, it decreases with the increase of the fluences. The lowest dielectric constant of PI irradiated by 1 MeV electron with 2.2 × 10^16^ e/cm^2^ is ~2.7. The dielectric constant of polyimide decreases with the increase of irradiation fluences in the whole frequency range. It is worth noting that the degree of attenuation of the dielectric constant is not linear with the increase in irradiation fluences as shown in Figure 1e. All samples show normal dielectric dispersion behavior, and the dielectric constant is only slightly reduced. All of the dielectric constant curves are frequency dependent. In the higher frequency range (>10^6^ Hz), the decrease in dielectric constant is more obvious. Under such high frequency conditions, the polar group with such orientation polarization motion cannot keep up with the change of electric field frequency, so the dielectric constant decreases.

Figure 1b,c shows the variation of the dielectric constant of PI samples before and after 3 MeV proton irradiation with different fluences (4.4 × 10^11^, 1.1 × 10^12^, and 2.2 × 10^12^ p/cm^2^) and 10 MeV proton irradiation with different fluences (5 × 10^11^, 1 × 10^12^, and 1 × 10^13^ p/cm^2^) as a function of the frequency at room temperature. Both dielectric constants of the irradiated PI decrease obviously with the increase of the fluences. All samples show normal dielectric dispersion behavior; the dielectric constant of PI samples decreases in the high-frequency range and the dielectric constant decreases with the increase of frequency. Figure 1d shows the variation of the dielectric constant of PI samples before and after 25 MeV carbon ion irradiated with different fluences (2.8 × 10^11^, 1.2 × 10^12^, and 5 × 10^12^ ions/cm^2^) as a function of the frequency at room temperature. The dielectric constant of polyimide decreases with the increase of irradiation fluences in the whole frequency range. However, the dielectric constant of irradiated polyimide begins to decrease at lower frequency when compared with the dielectric constant of the pristine PI, as the Figure 1d shown. The dielectric constant of pristine PI begins to decrease at 10^6^ Hz, the dielectric constant of irradiated PI with 2.8 × 10^11^ ions/cm^2^ begins to decrease at 10^5^ Hz, the dielectric constant of irradiated PI with 1.2 × 10^12^ ions/cm^2^ begins to decrease at 10^4^ Hz, and the dielectric constant of irradiated PI with 5 × 10^12^ ions/cm^2^ begins to decrease at 10^3^ Hz.

Figure 1e shows the variation in the dielectric constant of PI irradiated with different sources and fluences, the curve of 1 MeV electron uses bottom X-axis, and the curves of other irradiation sources use top X-axis at 100 Hz. The dielectric constant decreases nonlinearly with the increase of irradiation fluences. Figure 1f shows the measurement results with the rate of change of dielectric constant (at 100 Hz). The variation trend of Figure 4f is consistent with that of Figure 1e. These changes in the dielectric constant were calculated by using Equation (1).
(1)$$\nabla \varepsilon =\left(\varepsilon_{a}-\varepsilon_{b}\right)⁄\varepsilon_{b}\times 100\;\%$$

Here, *ε*_b_ is the dielectric constant of pristine PI and *ε*_a_ is the dielectric constant of irradiated PI.

According to the evolution characteristics of the dielectric properties of polymers, the dielectric constant of polymer is closely related to its characteristics of molecule, chain group structure, and the internal defects [21]. When the frequency is lower than 1 × 10^8^ Hz, the dielectric behavior of PI films mainly depends on the orientation polarization, the interface polarization, and dipole polarization, as shown in Figure 9. The interface polarization mainly comes from the defects after irradiation, as shown in Figure 8. The orientation polarization mainly comes from the main chain of molecule. This experiment is tested at room temperature, so the polarization is weak. The results are different from the high dielectric constant induced by irradiation such as gamma, so there may be some other main polarization mechanism leading to the reduction of dielectric constant here.

### 3.2. Dielectric Loss and Resistance Analysis

The dielectric loss of pristine and irradiated PI with different sources and fluences is shown in Figure 2 at different frequency and room temperature. From Figure 2a, it can be seen that the dielectric loss of the irradiated PI with 1 MeV electron increases compared with the pristine sample, especially at low frequency, but the dielectric loss increases with the increase in frequency (>10^4^ Hz) and the curves are almost the same after irradiation with different fluences. From Figure 2b,c, the data shows that the dielectric loss decreases a little with the increase in the frequency in the low frequency region, but the dielectric loss increases obviously with the increase of frequency in the high frequency region. It can be seen that the dielectric loss of the irradiated PI with 3 MeV proton shows a movement in the peak compared with the pristine sample at 10 Hz. The peak moves to 100 Hz after the 10 MeV proton. The dielectric loss of irradiated PI increases slightly compared with the pristine PI in the whole frequency. Figure 2d shows the variation of dielectric loss of PI irradiated by 25 MeV carbon ion in different frequency. The dielectric loss of PI irradiated by 25 MeV carbon ion increases with the increase of fluences in low frequency but the curves almost the same in high frequency compared with the pristine sample. The dielectric loss increases with the increase of the frequency for all irradiated and pristine PI samples. The dielectric loss depends on a number of factors, such as orientation polarization and dipole polarization. At low frequency, the dielectric loss is mainly caused by the main chain of the molecule movement polarization relaxation. There is a peak around 100 Hz that may be caused by the space charge polarization relaxation. The strong polarity in the molecular chain can produce unsaturated groups, such as carbonyl group, which have the polarization characteristics of space charge due to the movement ability limited, so the increase in the dielectric loss produces the peak. At high frequency, the dipole polarization relaxation produces the dielectric loss, and the dipole polarization cannot keep up with the change of frequency, thus the dielectric loss increase with the increase of the frequency.

The variation insulation resistance curves of PI irradiated with different fluences of four irradiation sources are shown in Figure 3 at the voltages of 500 V and 4000 V. It can be seen from the figure that the insulation resistance of PI decreases with the increase of fluences under different irradiation conditions. This indicates that all kinds of irradiation sources lead to the poor insulation of polyimide. This implies that the conductivity increases approximately with the increase of fluences. However, the resistance of the irradiated polyimide is 10^12^ Ω, which still maintains good insulation performance.

### 3.3. XPS Analysis

To investigate the chemical bond states and content of PI samples after irradiation, the XPS fine spectrum of pristine polyimide C1 s and O1 s and the detailed relative component ratios of different bonds irradiated by 3 MeV proton with 2.2 × 10^12^ p/cm^2^ and 25 MeV carbon ion with 5 × 10^12^ ions/cm^2^ are studied, as shown in Figure 4. The two peaks at 286.3 eV and 288.4 eV are attributed to C–O bonds and carbonyl group C=O bonds, respectively. By analyzing the content of different chemical bonds in the C1 s fine spectrum of XPS, it can reflect the structural damage behavior of polyimide after irradiation. Compared with the pristine PI, the relative content of C=O bonds and C–N bonds reduces after irradiation by 3 MeV proton and 25 MeV carbon ion, whereas the relative content of C–C bonds and C–O bonds increases, as shown in Table 1. Moreover, the content of C–C bonds changes a little after 3 MeV proton irradiation. However, the content of C–C bonds increases significantly after 25 MeV carbon ion irradiation, because the carbon ion irradiation produces a large number of pyrolytic carbon free radicals, and then the compound of these free radicals generates a lot of C–C bonds. Compared with the pristine PI, the relative content of C=O bonds decreases after irradiation by the 3 MeV proton and 25 MeV carbon ion, whereas the relative content of C–O bonds increases in the O1 s spectrum, as shown in Table 2. Thus, the content of polar groups such as C–N bonds and C=O bonds decreases, whereas the content of nonpolar groups such as C–C bonds increases after irradiation. This reduces the dielectric constant of the material, and forms a PI film with a low dielectric constant.

### 3.4. FT-IR Analysis

The FT-IR spectra of the pristine and irradiated PI films with different sources and fluences as a function of the frequency are shown in Figure 5. All of the samples exhibit typical characteristic imide peaks at 1780 cm^−1^ (imide C=O asymmetric stretching), 1720 cm^−1^ (imide C=O symmetric stretching), 1246 cm^−1^ and 1168 cm^−1^ (aromatic ether (C–O–C) asymmetric stretching vibration), and 1380 cm^−1^ (C–N–C stretching vibration). The intensity of the bands at 1500 and 1116 cm^−1^ represents the C=C stretching vibration and C–C stretching vibration, respectively [21]. The peak positions and patterns of the absorption peaks of the irradiated characteristic groups are similar to the spectral lines of the pristine samples, and no new characteristic absorption peaks generate, which indicates that no new groups generate inside the irradiated materials. By comparing the strength of the characteristic absorption peaks of materials before and after the four irradiation sources, it can be seen that the strength of characteristic absorption peaks of each group decreases slightly with the increase of irradiation fluences. The reason for this change may be the increase of material surface roughness caused by irradiation [22], this improves the material surface infrared scattering and weakens the intensity of absorption.

### 3.5. EPR Analysis

The EPR spectra of the pristine and irradiated PI films with different sources and fluences are shown in Figure 6: (a) 1 MeV electron, (b) annealing curves after 1 MeV electron, (c) 25 MeV carbon ion, and (d) the change of the content of free radicals. The results show that a large number of free radicals appear in the PI films under the condition of charged particles irradiation. In Figure 6a, the content of free radicals increases with the increase in the electron irradiation fluences. The content of free radicals changes greatly after carbon ion irradiation, as shown in Figure 6c. In Figure 6b,d, the content of free radicals after 1 MeV electron irradiation decreases as the change of time, which corresponds to the results of XPS data. This means that the free radicals on the surface from irradiated PI films recombine with the surrounding environment active factors such as oxygen from air, which leads to the decreasing of the content of free radicals. Figure 6d shows that the content of free radicals changes with the change of fluences or time after irradiation. The data are calculated by Formula (2):(2)N=∫∫Sdsm
where *N* is the content of free radicals, *S* is the measured EPR spectrum, and m is the quality of test sample here. As it can be seen from Figure 6d, the PI films produce a lot of free radicals and the content of the free radicals increases with the increase in the electron and carbon ions irradiation fluences. Moreover, more free radicals are produced after carbon ion irradiation than after electron irradiation. The g value is 2.0025 in this data, and the g value is stable and did not change with the irradiated particle energy and fluences. In the polymer material, the g values correspond to two kinds of free radical: one is a kind of pyrolytic carbon free radical, and the other is a hydroxyl superoxide radical. The irradiation experiment is under a vacuum environment, and therefore there is not enough oxygen reacting with free radicals, so it is impossible to generate such a large number of hydroxyl superoxide radicals. Moreover, combining the result in Figure 6c, the free radicals measured in experiment should be mainly pyrolytic carbon free radicals. The reason why the content of the free radicals after electron irradiation reduces in Figure 6d mainly because the free radicals on the surface recombine with active factors in the air, such as oxygen, as shown in Figure 8.

### 3.6. Mechanical Property Analysis

As we all know, stress is defined as the force per unit area of the samples and strain is the measure of the change in the sample length. The pristine and irradiated PI films with different sources and fluences exhibit typical stress–strain curves at room temperature, as shown in Figure 7. In the first stage of deformation, the stress gradually increases and reaches the yield strength about 110 Mpa, which shows the strain strengthening characteristics. With the obvious strain softening characteristics, it enters the second stage of deformation directly, and the tensile stress increases until the material breaks up. It is worth noting that the yield strength decreases slightly after the irradiation of four different sources, but the elongation at the break, at which the strain is ~60%, did not show an obvious change after irradiation. However, the elongation at the break decreased after the 10 MeV proton irradiation. The reason for this may be the degradation of PI after radiation, and then the reduction in molecular weight, resulting in the reduction in yield strength. Compared with other methods of introducing holes, which led to the mechanical properties of the materials decreasing sharply, the mechanical properties of the materials can still be maintained by the irradiation method. However, the disadvantage of this method is that a large number of fluences of irradiation for the material cannot be use; otherwise, the materials will degrade a lot and lose their available properties.

## 4. Discussion

The schematic of the irradiation process and irradiation degradation process and results of XPS data of PI repeat unit is shown in Figure 8. The schematic shows that the PI films irradiated by the four kinds of irradiation sources, including 1 MeV electron, 3 MeV proton, 10 MeV proton, and 25 MeV carbon ion, may produce defects, and the free radicals on the surface of the irradiated PI films and the free radicals may combine with active factors such as oxygen. The accelerated 1 MeV electron, 3 MeV proton, 10 MeV proton, and 25 MeV carbon ion all have enough energy to break chemical bonds in organic materials. The irradiation sources may break the benzene ring, ether bonds, C–N bonds, and C=O bonds of the PI films. The most common result of chemical bond breakage is the formation of free radicals. From the EPR results, we know that the free radicals are the pyrolytic carbon free radicals. The irradiation processes can be classified according to the effects of formation of free radicals, including curing, cross-linking, degradation, and grafting [24]. The experimental XPS results show that the content of C=O bonds and C–N bonds decreases and the content of C–O bonds and C–C bonds increases. According to the results, we can know degradation is the main process and the fracture of chemical bonds often forms polymers with low molecular weight.

The sample schematic of orientation polarization, interface polarization, and dipole polarization in the PI films is shown in Figure 9. The dielectric constant section refers to these polarizations. The orientation polarization mainly regards the molecular main chain polarization and the ability of the polarization relaxation related to the movement of main chain molecules. The reason why the interface polarization happens is that the irradiated PI films produce defects on the surface [24]. The dipole polarization is closely related to the relaxation process of the corresponding groups within the molecular features, and highly polarized groups in the molecular chain of polyimide, which can dominate this dipole polarization process. Table 3 and Table 4 show the typical molecular dipole moment and polarizability of the functional groups of the polyimide [25]. It can be seen that the groups C=O, C–O, and C–N have high polarizability and dipole moment, so electron density is concentrated around carbon, whereas oxygen and nitrogen are depleted in the condition of external field; these groups have strong polarization and corresponding relaxation phenomenon.

The dielectric constant of all samples decreases with the increase in the fluences and the rate of the decrease is not the same as all samples. Some researches have reported that polymer electrolyte films used electron beam and Gamma ray at 50 and 150 kGy, and the results showed that the dielectric constant increased with the increase of irradiation fluences and temperature because of the presence of an appreciable number of defects and dipoles in the form of chain scission [21]. Different irradiation sources, fluences, and irradiation rates have different effects on the structure, defects, and free radical damage of materials. The four different kinds of irradiation sources may have different mechanisms for low dielectric constant. The reasons why the four types of irradiation cause the decrease of dielectric constant are as follows. First, the molecular structure change is due to the increase of the free volume of the molecular structure and decrease of the density of the material. The large free volume reduces the number of polarizing groups per unit volume which can lead to the decrease of dielectric constant of the material. This should be the main reason for the decrease of the dielectric constant. Second, the irradiation may lead to defects of the PI films and the dielectric constant of air is 1, so the dielectric constant decreases. Third, the dipole polarization of the irradiated PI films decreases. The irradiation effect will directly lead to damage of the functional groups of the polymer. The polarity with large groups such as C=O, N–H, O–H, C–N, etc. is probably detached from the molecular main chain. Therefore, the decrease in the polarization ability of the groups inside the polymer material leads to the decrease in the dielectric constant.

## 5. Conclusions

The electron/proton/carbon ion beams of PI samples with different fluences have been systematically investigated, and the electrical and mechanical properties of the irradiated PI have been closely observed. The results show that the dielectric constant of PI irradiated by the four kinds of irradiation sources decrease with the increase of irradiation fluences. The dielectric loss of irradiated PI with the different fluences does not change obviously, but it increases significantly with the increase of frequency after 10^4^ Hz. The XPS data implies that the polytetrachloric anhydride group in the polyimide degrades after irradiation, the content of polar groups such as C–N bonds and C=O bonds decreases, whereas the content of non-polar groups such as C–C bonds increases. The results of the EPR show that irradiation produces a lot of pyrolytic carbon free radicals which will compound when the irradiated PI is putted in air. The mechanical properties decrease slightly after irradiation.

## Figures and Tables

**Figure 1 polymers-12-00879-f001:**
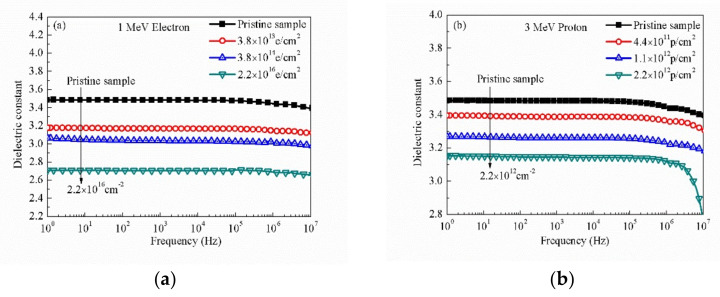
Frequency dependent dielectric constant of pristine and irradiated polyimide (PI) films with different sources and fluences at room temperature: (**a**) 1 MeV electron, (**b**) 3 MeV proton, (**c**) 10 MeV proton, (**d**) 25 MeV carbon ion, (**e**) dielectric constant at 100 Hz, and (**f**) the rate of change of dielectric constant at 100 Hz.

**Figure 2 polymers-12-00879-f002:**
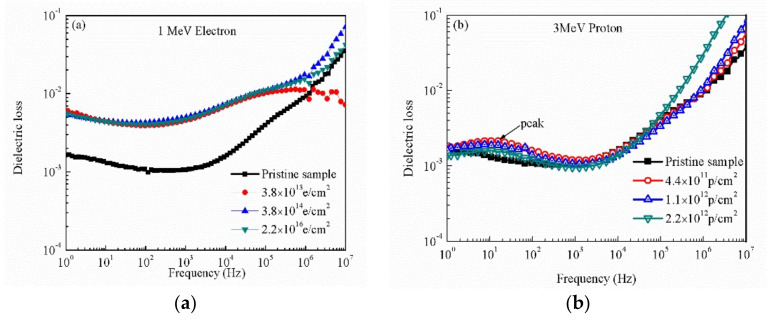
Frequency dependent dielectric loss of pristine and irradiated PI films with different sources and fluences at room temperature: (**a**) 1 MeV electron, (**b**) 3 MeV proton, (**c**) 10 MeV proton, and (**d**) 25 MeV carbon ion.

**Figure 3 polymers-12-00879-f003:**
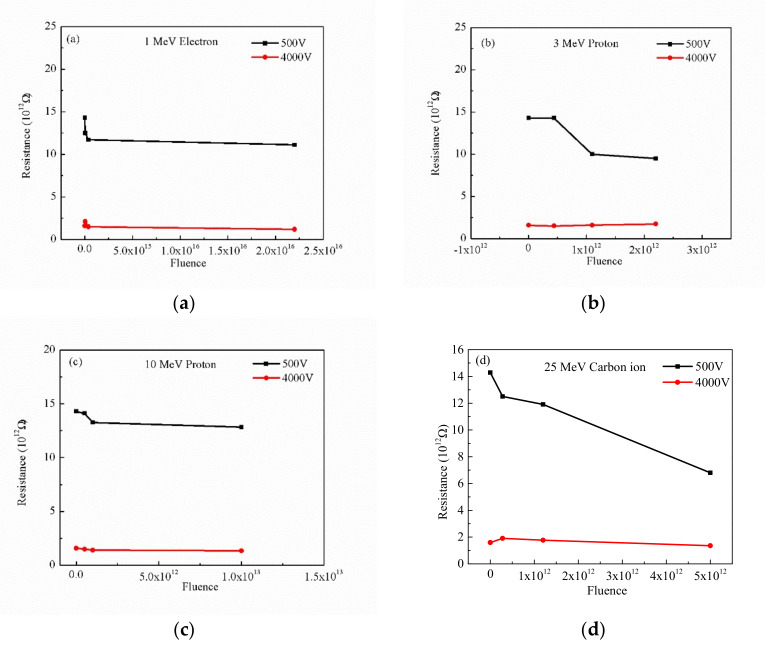
Frequency dependent resistance of pristine and irradiated PI films with different sources and fluences at room temperature: (**a**) 1 MeV electron, (**b**) 3 MeV proton, (**c**) 10 MeV proton, and (**d**) 25 MeV carbon ion.

**Figure 4 polymers-12-00879-f004:**
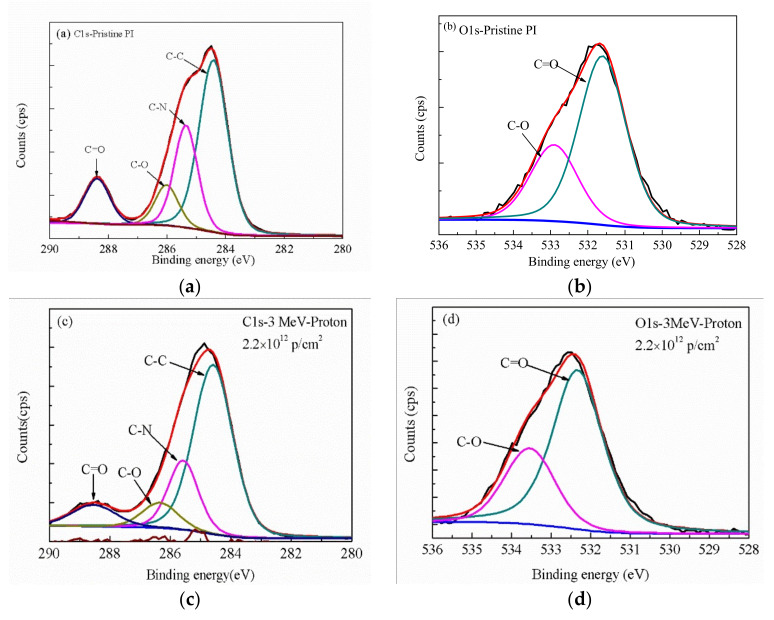
X-ray photoelectron spectroscopy (XPS) curves of pristine and irradiated PI films: (**a**) C1 s of pristine PI, (**b**) O1 s of pristine PI, (**c**) C1 s of irradiated PI with 3 MeV proton of 2.2 × 10^12^ p/cm^2^, (**d**) O1 s of irradiated PI with 3 MeV proton of 2.2 × 10^12^ p/cm^2^, (**e**) C1 s of irradiated PI with 25 MeV carbon ion of 5 × 10^12^ p/cm^2^, and (**f**) O1 s of irradiated PI with 25 MeV carbon ion of 5 × 10^12^ p/cm^2^.

**Figure 5 polymers-12-00879-f005:**
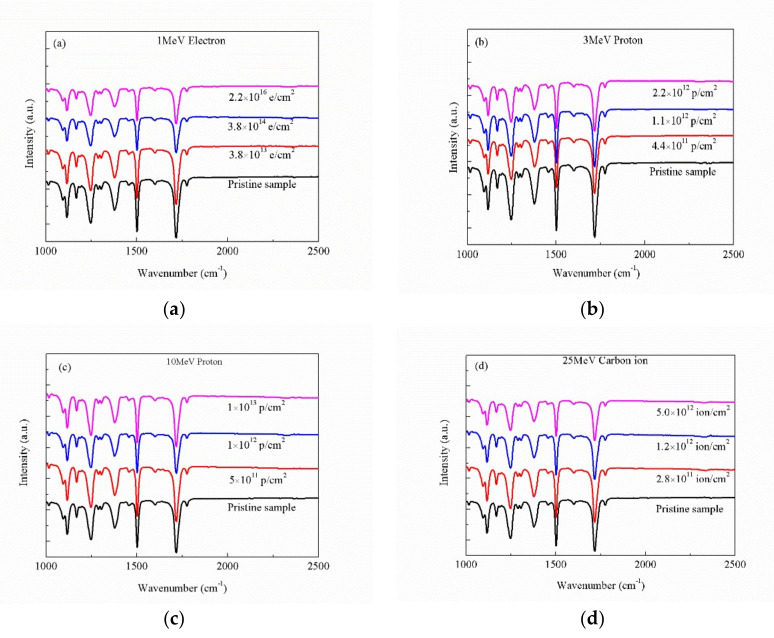
FT-IR curves of pristine and irradiated PI films with different sources and fluences: (**a**) 1 MeV electron, (**b**) 3 MeV proton, (**c**) 10 MeV proton, and (**d**) 25 MeV carbon ion.

**Figure 6 polymers-12-00879-f006:**
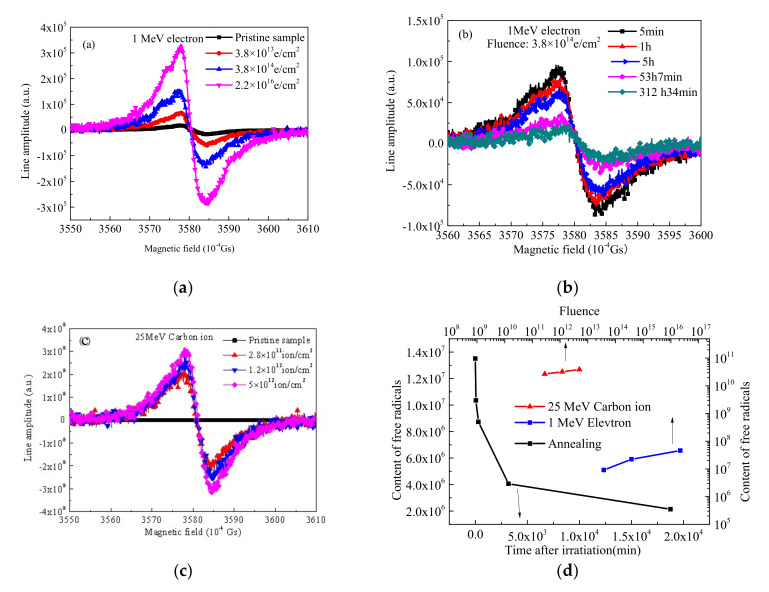
Electron paramagnetic resonance (EPR) curves of the pristine and irradiated PI films with different sources and fluences: (**a**) 1 MeV electron, (**b**) annealing after 1 MeV electron, (**c**) 25 MeV carbon ion, and (**d**) the change of the content of free radicals.

**Figure 7 polymers-12-00879-f007:**
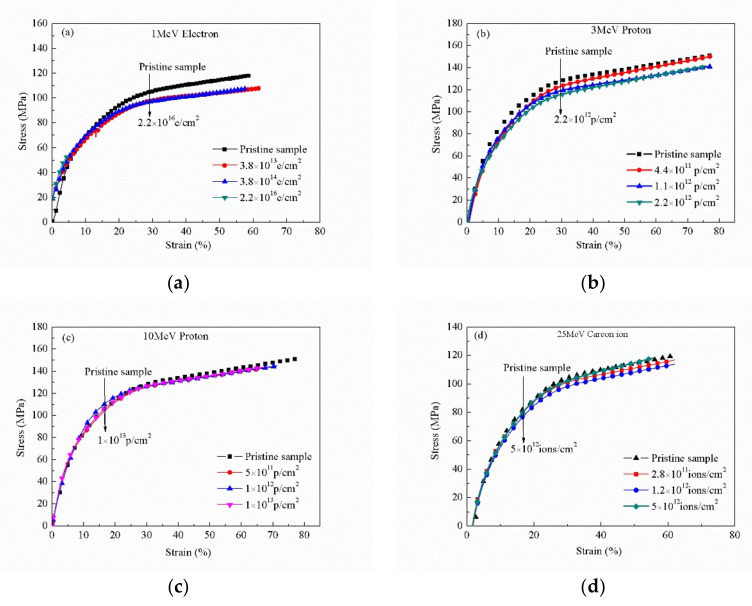
Stress–strain curves of pristine and irradiated PI films with different sources and fluences at room temperature: (**a**) 1 MeV electron, (**b**) 3 MeV proton, (**c**) 10 MeV proton, and (**d**) 25 MeV carbon ion.

**Figure 8 polymers-12-00879-f008:**
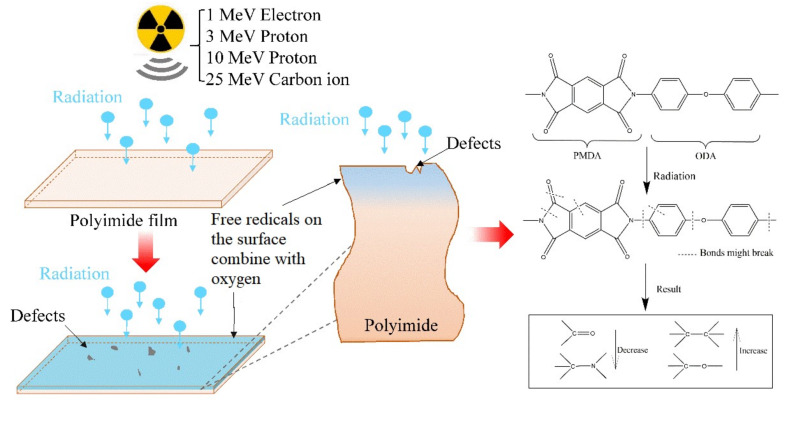
Schematic of the irradiation process and irradiation degradation process of PI repeat unit and the XPS results show that the content of C=O and C–N decrease and the content of C–C and C–O increase.

**Figure 9 polymers-12-00879-f009:**
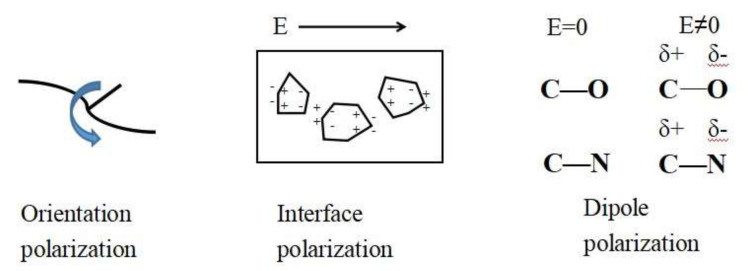
Simple schematic of orientation polarization, interface polarization and dipole polarization happened in PI films.

**Table 1 polymers-12-00879-t001:** C element composition of PI with irradiation energy of 3 MeV proton and 25 MeV carbon ion.

Irradiation Energy	Fluence (cm^−2^)	Chemical Bond	Binding Energy (eV)	Proportion (%)
Pristine Polyimide		C–C	284.5	60.4
		C–N	285.5	20.1
		C–O	286.3	6.8
		C=O	288.4	12.7
3 MeV Proton	2.2 × 10^12^	C–C	284.5	65.7
		C–N	285.5	16.8
		C–O	286.5	7.5
		C=O	288.5	10.0
25 MeV Carbon ion 5.0 × 10^12^		C–C	284.5	71.6
		C–N	285.5	11.4
		C–O	286.5	7.3
		C=O	288.5	9.7

**Table 2 polymers-12-00879-t002:** O element composition of PI with irradiation energy of 3 MeV proton and 25 MeV carbon ion.

Irradiation Energy	Fluence (cm^−2^)	Chemical Bond	Binding Energy (eV)	Proportion (%)
Pristine Polyimide		C=O	532	74.1
		C–O	533.2	25.9
3 MeV Proton	2.2 × 10^12^	C=O	532	71.1
		C–O	533.2	28.9
25 MeV Carbon ion	5.0 × 10^12^	C=O	532	70.6
		C–O	533.2	29.4

**Table 3 polymers-12-00879-t003:** Polarizabilities of the chemical bonds parallel and normal to the bond axis and the mean value for all three directions in space [24].

Bond	α_//_	α_Ʇ_	α_m_
C–N	0.58	0.84	0.75
C–H	0.79	0.58	0.65
C–C	1.88	0.02	0.64
C–O	2.25	0.48	1.07
C=O	2.00	0.75	1.20

**Table 4 polymers-12-00879-t004:** Dipole moments of the chemical bonds of the relevant molecules expressed in Debye units [24].

Bond	μ (10^−30^/CM)
C–N	0.40
C–H	0.74
C–C	0.00
C–O	2.30
C=O	0.73

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
