# Peer review of "Low Dielectric Constant Polyimide Obtained by Four Kinds of Irradiation Sources"

_polymers, 2020, doi:10.3390/polym12040879_

Round 1
Reviewer 1 Report
The manuscript presents study of mechanical, electric and structural properties of polyimide (PI) films irradiated with high-energy electrons, protons and carbon atoms. Based on the obtained results, the authors concluded that the dielectric constant of irradiated PIs decreases with irradiation. The presentation has many English flaws and misprints. It is not ready for publishing in the present form. I have few critical points below.
Comments
Major
Your main conclusion is based on the observation that amount of C=O bonds decreases after irradiation with protons and carbon atoms (line 293-297). It is confusing. First, particle irradiation creates defects and bond cleavage, thus, electric dipole of unsaturated bonds tends to increase the dielectric constant, such as reported in J Radiation Res & Appl Sci v.9, p. 117 (2016). Second, decrease of C=O bonds and increase of C-O bonds (line 295) must lead to increase in the dielectric constant because C-O has very large dipole moment compared to others. Therefore, this conclusion is not supported by your data. Moreover, I suspect that the air in the contacts between aluminum electrodes and PI films have to be responsible for lowering of the dielectric constant. You did not give any information on the details of the measurements. Therefore, the presented data are doubtful. The reduction of the dielectric constant by irradiation must be confirmed by ellipsometry. Your discussion on the mechanism is abridged. There is substantial difference in interaction with material between particles used -electrons, protons, carbon atoms. For example, protons can create voids, which electrons do not. You oversimplified your conclusion. You must discuss it separately. Line 49-50, “irradiation is a new method to obtain low dielectric constant polymer.” The statement is incorrect. There are many applications of high-energy particle irradiation, such as dopant implantation and electron beam resist processing methods are common in microelectronic industry. Also, particle irradiation of polymers has been known for about 50 years. Details on the measured techniques are missing. What are irradiation area and particle flux? How the stopping power (i.e. a depth at which the particles lost all kinetic energy) compares with the film thickness? How did you attach the electrodes? Load force? What technique did you use to measure dielectric constant – electrical or optical? The manuscript is poorly organized.The design of the experiment (Fig.7 and text) is given at the end, and tensile stress result in the beginning, though, it is not main subject.
The discussion is missing specific values to compare. Such as, there are no mechanical values given to the yield strength and elongation at the break (line 99, 103), no indication on the Fig 1 . Did you measure "engineering stress" in Fig.1?
The discussion on dielectric loss and electric polarization (around line 202) are indistinct. Eq(2) is a common knowledge. Did you use it in your analysis? Have you made an estimate of number of dipoles from your films, permittivity value ? Rewrite or remove.
There are misprints in reference list, such as 1, 9 and 24 are incomplete.
7. Technical points
There are incorrect English sentences. For example, line 26-29, line 44, 54, 65-66, 88, 268-269 and others.
A wording “fluences change with frequency” was used inadequately in many places.
Numbers in the text are given in plain format, no superscripts.
Line 128, “increase of material surface roughness” need a reference. …
Author Response
Dear Editor,
Thank you very much for your reply and help. Thanks a lot for the reviewers’ comments and their kind suggestions of our manuscript (Polymers-717393) entitled “Low Dielectric constant Polyimide obtained by Irradiation” changed “Low Dielectric constant Polyimide obtained by Four kinds of Irradiation sources”We have revised the article as the reviewers’ suggestions. In order to make the changes easily viewable for you and reviewers, in the revised manuscript, we marked the revisions with red color. We have checked carefully every sentences of the whole manuscript and corrected the grammar mistakes. We hope the revised paper would satisfy you and the reviewers.
We are looking forward to hearing from you soon.
Kind Regards,
Xingji li
..........................................................................................................................................
Revision list according to the comments
Reviewer :1.Your main conclusion is based on the observation that amount of C=O bonds decreases after irradiation with protons and carbon atoms (line 293-297). It is confusing. First, particle irradiation creates defects and bond cleavage, thus, electric dipole of unsaturated bonds tends to increase the dielectric constant, such as reported in Jour. of Radiation Res.& Applied Sci. volume.9, page. 117 (2016). Second, decrease of C=O bonds and increase of C-O bonds (line 295) must lead to increase in the dielectric constant because C-O has very large dipole moment compared to others. Therefore, this conclusion is not supported by your data. Moreover, I suspect that the air in the contacts between aluminum electrodes and PI films have to be responsible for lowering of the dielectric constant. You did not give any information on the details of the measurements. Therefore, the presented data are doubtful. The reduction of the dielectric constant by irradiation must be confirmed by ellipsometry.
Reply:
This is a good question, we also ever doubt the conclusion, indeed gamma irradiation can increase the dielectric constant of the polymer material as the paper reported, we also got such results in PI films. You know polymer irradiation exist degradation and crosslinking at same time, the degree of degradation and crosslinking for different irradiation source and irradiation fluence are different, so low dielectric polymer obtained by irradiation is possible. This experiment adopt the four kinds of radiation sources are different with gamma, we get the low dielectric constant of the PI and repeat the experiment also get the same results, otherwise, the four radiation sources all get similar low dielectric constant result. In fact, we want to get the conclusion that irradiation is a method to get low dielectric constant materials when the content of polar group decrease. We agree with your point of view that its insufficient evidence to get the conclusion. Therefore, we changed the article and just report that the four kinds of irradiation sources were used to irradiate the PI films than obtaining the low dielectric constant of PI films. So the title change to “Low Dielectric constant Polyimide obtained by Four kinds of Irradiation sources” which is more appropriate for this article. The reason for decrease of C=O bonds and increase of C-O bonds( line 295) is that the decreament of C=O bonds and increment of C-O bonds are few, and content of C-C increase a lot. Therefore, the total polarization rate decreased. We rewrite it, we copy some here for your check.
The fourth is that due to irradiation cause the decrease of the dipole polarization in the whole frequency range at room temperature. For the 1 MeV electron irradiation maybe due to the content of non-polarized group increase, irradiation damage lead to the number of the typical dipole groups of C=O bonds and C-N bonds decrease resulting in a decline of the dipole polarization in polyimide. Even though the C-O bonds has large dipole moment, according to the result of XPS table 1 and table 2, the content of C-O increase very little, but the content of C-C the increase large and the content of C=O bonds and C-N bonds decrease, so the overall result shows that the dielectric constant of polyimide decreased. In the case of irradiation, when the contents of non-polar groups increase, the dielectric constant of the material decreases.
Your discussion on the mechanism is abridged.There is substantial difference in interaction with material between particles used -electrons, protons, carbon atoms. For example, protons can create voids, which electrons do not.
Reply:
Thank you for your advice. We have add the mechanism in part of the discussion which is discussed separately and we copy here for your check.
The different irradiation source have different mechanisms for low dielectric constant. The four irradiation causes the dielectric constant to decrease due to the following four reasons. Firstly, degradation of the molecular structure increases the free volume of the molecular structure and reduces the density of the material thus making the dielectric constant of the material decrease. Secondly, for the proton irradiation maybe due to the PI films produce voids, and the dielectric constant of air is 1, so the dielectric constant decreases. Thirdly, the reason the carbon ion maybe due to the polymer form three-dimensional network structure, the chain movement ability become weak and the decrease of orientation polarization. The fourth is that due to irradiation cause the decrease of the dipole polarization in the whole frequency range at room temperature. For the 1 MeV electron irradiation maybe due to the content of non-polarized group increase, irradiation damage lead to the number of the typical dipole groups of C=O bonds and C-N bonds decrease resulting in a decline of the dipole polarization in polyimide. Even though the C-O bonds has large dipole moment, according to the result of XPS table 1 and table 2, the content of C-O increase very little, but the content of C-C the increase large and the content of C=O bonds and C-N bonds decrease, so the overall result shows that the dielectric constant of polyimide decreased. In the case of irradiation, when the contents of non-polar groups increase, the dielectric constant of the material decreases.
You oversimplified your conclusion. You must discuss it separately. Line 49-50, “irradiation is a new method to obtain low dielectric constant polymer.” The statement is incorrect. There are many applications of high-energy particle irradiation, such as dopant implantation and electron beam resist processing methods are common in microelectronic industry. Also, particle irradiation of polymers has been known for about 50 years. Details on the measured techniques are missing. What are irradiation area and particle flux? How the stopping power (i.e. a depth at which the particles lost all kinetic energy) compares with the film thickness?
Reply:
Thank you for your remind. We have deleted the sentence(line 49-50), change the view and report that the four kinds of irradiation sources were used to irradiate the PI films than obtaining the low dielectric constant of PI films. Otherwise, we change the aritcle title as the question 1 I answer you. The details on the measured techniques include irradiation area and particle flux have been added in 2.2 Experimental parameter. The stopping power (i.e. a depth at which the particles lost all kinetic energy) compares with the film thickness have been calculated by Geant4 and SRIM and add in 2.2 Experimental parameter. We copy here for your check.
The PI films were cutted into 5×5 cm size and all the irradiation area are larger than this size. The electron energy is 1 MeV, particle radiation flux is 2×108 cm-2·s-1, the irradiation fluences are 3.8×1013, 3.8×1014 and 2.2×1016 e/cm2, respectively. The 1 MeV electron penetration depth is about 100 μm calculated by Geant4 (GEometry ANd Tracking). The proton energy is 3 MeV, the 3 MeV proton penetration depth is about 100 μm calculated by SRIM (Stopping and Ranges of Ions in Matter) software and the irradiation fluences are 4.4×1011, 1.1×1012 and 2.2×1012 p/cm2, respectively. The proton energy is 10 MeV, the 10 MeV proton penetration depth is about 1000 μm calculated by SRIM software and the irradiation fluences are 5×1011, 1×1012 and 1×1013 p/cm2, respectively. The carbon ion energy is 25 MeV, the 25 MeV carbon ion penetration depth is about 30 μm calculated by SRIM software, the films thickness is 50 μm, thus the 25 MeV carbon ion not penetrate though the PI films and the irradiation fluences are 2.8×1011, 1.2×1012 and 5×1012 ion/cm2, respectively.
How did you attach the electrodes? Load force? What technique did you use to measure dielectric constant – electrical or optical?
Reply:
Thank you for your remind. We had added the method and technology in 2.3. Characterizations. We copy here for your check.
The dielectric constant, dielectric loss and insulation resistance of the samples were measured with BDS 4000 Novocontrol broadband dielectric spectrometer/impedance produced in Germany in a wide frequency range from 10-1 to 107 Hz at room temperature. Both sides of the samples before test evaporated a 25 mm diameter Al electrode by the vacuum evaporation machine. The equipment used in external stimulus is an electric field E, the relative dielectric constant corresponds to the outside world and generate electricity the degree of polarization electric stimulation.
The manuscript is poorly organized.The design of the experiment (Fig.7 and text) is given at the end, and tensile stress result in the beginning, though, it is not main subject. The discussion is missing specific values to compare. Such as, there are no mechanical values given to the yield strength and elongation at the break (line 99, 103), no indication on the Fig 1 . Did you measure "engineering stress" in Fig.1? The discussion on dielectric loss and electric polarization (around line 202) are indistinct. Eq(2) is a common knowledge. Did you use it in your analysis? Have you made an estimate of number of dipoles from your films, permittivity value ? Rewrite or remove.
Reply:
Thank you for your suggest. We have reorganized the article, put the dielectric constant on the front and stress-strain on the last which add the specific values to compare on the back. Moreover, the EPR data an analysis was add to the paper for explaining the changes of free radicals after irradiation . See the revised manuscript. Otherwise, we add the specific values and remove the common knowledge Eq(2). We copy here for your check.
In the first stage of deformation, the stress gradually increases and reaches the yield strength about 110 Mpa, showing the strain strengthening characteristics.
It is worth noting that the yield strength slightly decreases after irradiation for the four different sources, but the elongation at the break which the strain is about 60% did not show an obvious change after irradiation.
In dipole polymer groups, under the effect of outfield turning-direction polarization, the dipole polarization is mainly produced in the material group of dipole vibration. The vibration frequency is higher than that in the main chain of the molecule polarization, so the frequency range of the dipole relaxation is a little bit higher than that of the main chain polarization. However, the dielectric constant of polyimide is mainly contributed by dipole polarization at room temperature[24].
3.5. EPR analysis
The EPR spectra of pristine and irradiated PI films with different sources and fluences are show in Figure 6, (a) 1 MeV electron, (b) annealing curves after 1 MeV electron, (c) 3MeV proton, (d) 25MeV carbon ion. The results show that there are a large number of free radicals generated in PI films under the condition of charged particles irradiation. The content of free radicals increase with the electron irradiation fluerence increase from the figure(a). The content of free radical change a lot after carbon ion irradiation as the Figure 6(d) shown. The content of free radicals after 1 MeV electron irradiation decreased as the change of time are shown in Figure 6(b). This correspond to the results of XPS. This means that its compound between the free radicals and the surrounding environment active factors such as oxygen, leading to the content of free radicals continues to reduce. The g value is 2.0025 in this data, and the g value is stable and did not change with the irradiated particle energy and fluence. In polymer material, the g values corresponding to two kinds of free radical, one is a kind of pyrolytic carbon free radicals, the other is a hydrocarbyl super oxygen free radicals. During the irradiation experiment is a vacuum environment, there is not enough oxygen react with free radicals, so it’s impossible to generate such a large number of hydroxyl oxygen free radicals. Moreover combining the result figure(d) the free radicals mainly measured in experiment should be pyrolytic carbon free radicals. When the material form a large number of C-C bonds, the dielectric constant of the PI films will decrease, and it corresponding dielectric constant experimental results.
|
|
|
|
(a) |
(b) |
|
|
|
|
(c) |
(d) |
Figure 6. EPR curves of pristine and irradiated PI films with different source and fluences (a)1MeV electron, (b) annealing after 1MeV electron, (c)3MeV proton, (d) 25MeV carbon ion.
There are misprints in reference list, such as 1, 9 and 24 are incomplete.
Reply:
Thank you for your remind. We have complete the reference and check all the reference. We copy here for your check.
Vesely, D. S. Finch and G. E. Cooley. Electrical properties of polymers modified by electron beam irradiation. Polymer,August 1988, Pages 1402-1406. Neese, B. Chu, S. Lu, et al., Large Electrocaloric Effect in Ferroelectric Polymers Near Room Temperature. Science, 2008, Vol.3211(5890):821-3. Yue, Investigation on charge particle radiation-induced conductivity and dielectric properties of polyimide,Nuclear Instruments and Methods in Physics Research Section B: Beam Interactions with Materials and Atoms. Volume 291, 15 November 2012, Pages 17-21. There are incorrect English sentences. For example, line 26-29, line 44, 54, 65-66, 88, 268-269 and others.A wording “fluences change with frequency” was used inadequately in many places.Numbers in the text are given in plain format, no superscripts.Line 128, “increase of material surface roughness” need a reference.Reply:
Thank you for your remind. We have modified the incorrent sentences, modified the “fluences change with frequency”, revised superscripts and subscripts and add the reference[25], check all the english grammar. We copy some here for your check.
In recent years, with the rapid development of microelectronics industry, the miniaturization of electronic components, the rapid growth of large scale integrated circuit chips, the increase of chip interconnect density, and the improvement the circuit connection of the resistance and capacitance, the signal delay of resistance production effect signal transmission speed and signal loss[1-3].
The reason for dielectric constant of the PI-PDMS polymer decrease is that polydimethylsiloxane (PDMS) was added to the polyimide than the chain reaction produces carbon dioxide to form nanoporous films.
Irradiation changes the structure and properties of polymer materials, the dielectric constant of the irradiated polymer decreased when the content of polarized group decrease.
The four kinds of Irradiation sources can decrease the dielectric constant of PI films when the content of non-polarized group increase.
The dielectric constant, dielectric loss and insulation resistance of the samples were measured with BDS 4000 Novocontrol broadband dielectric spectrometer/impedance produced in Germany in a wide frequency range from 10-1 to 107 Hz at room temperature.
Figure 1. Frequency dependent dielectric constant of pristine and irradiated PI films with different source and fluences at room temperature.
Figure 2. Frequency dependent dielectric loss of pristine and irradiated PI films with different source and fluences at room temperature.
Figure 3. Frequency dependent resistance of pristine and irradiated PI films with different source and fluences at room temperature.
J.Dutta,S.R.Mohanty, et al., Self-organized nanostructure formation on the graphite surface induced by helium ion irradiation. Physics Letters A. Volume 382, Issue 24, 19 June 2018, Pages 1601-1608.

Reviewer 2 Report
Paper by Li et al describes effect of irradiation on electric properties of polyimide film. There are several points those have to be clarified before publication:
Major points:
Infrared spectra: Authors give two possible explanations for decrease in intensity but they should give different changes. Increased toughness of the samples should result in proportional decrease of all bands (should be slightly wavelength dependent) but reduction of the function groups only for particular bands. From given figures, it is hard to decide, as no visible changes are observable in the infrared spectra. At given resolution, all of them seems to be the same. The first option is more probable as radiation influence mainly the surface of the sample while IR technique collects spectra of the whole sample profile. Changes that are more distinct could be probably obtained by ATR technique or by specular reflectance. XPS: The obtained spectra show significant differences caused by irradiation but quantification is hardly acceptable. The C=O/C–O ratio determined from C1s and O1s regions are very different. The C1s data of pristine PI are consistent with polymer structure given in Figure 7 but O1s strongly overestimate C=O. Further for XPS: Authors do not distinct between minor changes in chemical composition (e.g. content of C–C after proton irradiation) and major changes (e.g. C–C after C irradiation). Furthermore, such changes should be clarified. The earlier example is probably only experimental error. In the later example, it is very probably due to absorption of carbon from irradiation source. Irradiation with high-energy particles is often connected with polymer degradation and changes of polymer morphology but only few of common experimental methods are used. In my opinion, all samples should be characterized by DSC and by electron microscopy. EPR spectroscopy should be used for detection of radical species appearing upon irradiation. Such species are often stabilized with polymeric matrix; they can strongly influence electric properties of polymeric materials.Minor points:
Lines 70, 71: Please check use of subscripts and superscripts through the manuscript.
Line 70: 50 um?
Line 85: “spectrums” should read “spectra”
Figure 3b: Different color is used for C-O band than in the other figures.
In summary, the manuscript should not be published in current form. Major revision is necessary.
Author Response
Dear Editor,
Thank you very much for your reply and help. Thanks a lot for the reviewers’ comments and their kind suggestions of our manuscript (Polymers-717393) entitled “Low Dielectric constant Polyimide obtained by Irradiation” changed “Low Dielectric constant Polyimide obtained by Four kinds of Irradiation sources”We have revised the article as the reviewers’ suggestions. In order to make the changes easily viewable for you and reviewers, in the revised manuscript, we marked the revisions with red color. We have checked carefully every sentences of the whole manuscript and corrected the grammar mistakes. We hope the revised paper would satisfy you and the reviewers.
We are looking forward to hearing from you soon.
Kind Regards,
Xingji li
..........................................................................................................................................
Revision list according to the comments
Reviewer: 1.Infrared spectra: Authors give two possible explanations for decrease in intensity but they should give different changes. Increased toughness of the samples should result in proportional decrease of all bands (should be slightly wavelength dependent) but reduction of the function groups only for particular bands. From given figures, it is hard to decide, as no visible changes are observable in the infrared spectra. At given resolution, all of them seems to be the same. The first option is more probable as radiation influence mainly the surface of the sample while IR technique collects spectra of the whole sample profile. Changes that are more distinct could be probably obtained by ATR technique or by specular reflectance.
Reply:
Thank you for your advice. It is a good suggests to analysis the change of intensity. You know all companies delay to work due to the novel coronavirus in China, so far our school and many institutes still not start to work, thus we don't have enough conditions to test ATR technique or by specular reflectance. It’s pity we are unable to test this experimental data, we will adopt your suggestion for the next research paper, thank you very much for your understanding. Otherwise, The FT-IR spectrum we use to the article mainly analyze the group of the PI after irradiation, the results show that no new peak form and disappear, this means the group structure of the PI material after irradiation did not change. The results can support the interpretation of the dielectric constant.
XPS: The obtained spectra show significant differences caused by irradiation but quantification is hardly acceptable. The C=O/C–O ratio determined from C1s and O1s regions are very different. The C1s data of pristine PI are consistent with polymer structure given in Figure 7 but O1s strongly overestimate C=O. Further for XPS: Authors do not distinct between minor changes in chemical composition (e.g. content of C–C after proton irradiation) and major changes (e.g. C–C after C irradiation). Furthermore, such changes should be clarified. The earlier example is probably only experimental error. In the later example, it is very probably due to absorption of carbon from irradiation source. Irradiation with high-energy particles is often connected with polymer degradation and changes of polymer morphology but only few of common experimental methods are used. In my opinion, all samples should be characterized by DSC and by electron microscopy. EPR spectroscopy should be used for detection of radical species appearing upon irradiation. Such species are often stabilized with polymeric matrix; they can strongly influence electric properties of polymeric materials.
Reply:
Thank you for your suggests.We are very agree with your point of view. We think the experimental error is allow in the paper, the reference (Xin jianjiao, X-ray Photoelectron Spectroscopy study of Polyimide Film by Ar Ion Bombardment, Chemical Industy Times Vol 29No.1 Jan1.2015) also show the similar results about the XPS spectra. We already modified the distinct about minor and major changes. For the DSC and SEM, same reason as the reply 1,Of course it can't be an excuse. The SEM of PI films after another energy electron we have ever been measured, no obviously difference with the pristine one. Thus this paper we did not measure. The peak and crystallinity of DSC have a little change after irradition, this time we are so sorry that we can not add to this paper due to the limit of the condition,but its good suggests we will add to next research, thank you for your understanding.However, the EPR results we tested before just not add to the article, this time we add the EPR Characterizations and data to this paper. We copy here for your check.
Moveover, the content of C-C bonds change a little after 3MeV proton irradiation. However, the content of C-C bonds increased significantly after 25MeV carbon ion irradiation, due to the carbon ion irradiation produce a large number of pyrolytic carbon free radicals, after compound of this free radicls generated a lot of C-C bonds.
Free radicals were tested by the A200 type Electron paramagic resonance (EPR) from the German Bruke company, this experiment adopts the center of the magnetic field is 3580 G, sweep the field width is 90 G, microwave frequency is 9.85 GHz, time constant is 20 s, spectral gain is 2×104, modulation amplitude is 1 G and microwave power is 6.140 mW.
3.5. EPR analysis
The EPR spectra of pristine and irradiated PI films with different sources and fluences are show in Figure 6, (a) 1 MeV electron, (b) annealing curves after 1 MeV electron, (c) 3MeV proton, (d) 25MeV carbon ion. The results show that there are a large number of free radicals generated in PI films under the condition of charged particles irradiation. The content of free radicals increase with the electron irradiation fluerence increase from the figure(a). The content of free radical change a lot after carbon ion irradiation as the Figure 6(d) shown. The content of free radicals after 1 MeV electron irradiation decreased as the change of time are shown in Figure 6(b). This correspond to the results of XPS. This means that its compound between the free radicals and the surrounding environment active factors such as oxygen, leading to the content of free radicals continues to reduce. The g value is 2.0025 in this data, and the g value is stable and did not change with the irradiated particle energy and fluence. In polymer material, the g values corresponding to two kinds of free radical, one is a kind of pyrolytic carbon free radicals, the other is a hydrocarbyl super oxygen free radicals. During the irradiation experiment is a vacuum environment, there is not enough oxygen react with free radicals, so it’s impossible to generate such a large number of hydroxyl oxygen free radicals. Moreover combining the result figure(d) the free radicals mainly measured in experiment should be pyrolytic carbon free radicals. When the material form a large number of C-C bonds, the dielectric constant of the PI films will decrease, and it corresponding dielectric constant experimental results.
|
|
|
|
(a) |
(b) |
|
|
|
|
(c) |
(d) |
Figure 6. EPR curves of pristine and irradiated PI films with different source and fluences (a)1MeV electron, (b) annealing after 1MeV electron, (c)3MeV proton, (d) 25MeV carbon ion.
3.Lines 70, 71: Please check use of subscripts and superscripts through the manuscript.
Reply:
Thank you for your remind. We have modified all the subscripts and superscripts through the manuscript. We copy some here for your check.
The electron energy is 1 MeV, particle radiation flux is 2×108 cm-2·s-1, the irradiation fluences are 3.8×1013, 3.8×1014 and 2.2×1016 e/cm2, respectively. The 1 MeV electron penetration depth is about 100 μm calculated by Geant4 (GEometry ANd Tracking). The proton energy is 3 MeV, the 3 MeV proton penetration depth is about 100 μm calculated by SRIM (stopping and ranges of ions in matter) software and the irradiation fluences are 4.4×1011, 1.1×1012 and 2.2×1012 p/cm2, respectively.
Line 70: 50 um?Line 85: “spectrums” should read “spectra”Reply:
Thank you for your remind. We have modified it and check all the manuscript mistake. We copy here for your check.
Kapton-H polyimide (PI, Isophthalic anhydride diaminodiphenyl ether) films with the density of 1.34 g/cm3 and 50 μm in thickness purchased from Du Pont co., Ltd., USA, with molecular formula (C22H10N2O5)n.
Fourier infrared spectroscopy (FT-IR) spectra of films have been obtained by Nicolet Magna-IR 560 spectrometer operated in transmission mode with the spectral region between 4000 and 700 cm-1 and scanned 32 times on average.
Figure 3b: Different color is used for C-O band than in the other figures.In summary, the manuscript should not be published in current form. Major revision is necessary.
Reply:
Thank you for your remind. We have revised the color of the C-O to same with others. Thank you for your time to review this paper, and put forward valuable suggestions, we have revised the article carefully according to your opinion, hope it can satisfied you.

Round 2
Reviewer 1 Report
The authors have changed the manuscript. They added the experimental details, rearranged the figures, added new results by EPR, and made English corrections in some parts. However, the presentation remains ambiguous. The main conclusion – the decrease in the dielectric constant after particle irradiations is due to reduction of amount of polar bonds – is not supported by presented results. The description of the mechanism is vague, and analysis of experimental data is misleading and suffers with many inconsistent statements. Also, flaws in English make it difficult to understand the results. The manuscript does not give meaningful information in the present form, and needs significant modification.
Particular comments
- The major critique is that there is no physically meaningful analysis of data. The evidence of specific polar bond cleavage is not given. XPS data are probing the surface composition only, and it may differ from that in the film bulk. FTIR data are valuable for analysis, but there is no detail comparison at all. EPR data are puzzling as the peak shifts for 3MeV protons. It causes doubt in correctness of the EPR measurements. Moreover, amount of free radicals are the largest after carbon irradiation as expected, but the dielectric constant change is the smallest, and the implanted carbon atoms stay in the PI film. Therefore, the cleavage of bonds is unlikely be the main effect here. You should consider alternative explanations. Also, as I said early, analysis of the mechanism has to be discussed separately for each particle. In particular, large effect of 1MeV electron irradiation need to be discussed in details.
- The studied PI films (50 um) are transparent for high-energy electrons and protons, and therefore, they should break chemical bonds and create defects in the bulk. It contradicts the irradiation process in Fig.8 where the top layer is modified. Moreover, “oxide layer” in Fig.8 is inconsistent with your statement on page 10, line 250, and the XPS data did not show surface oxidation (only change of C-O by 0.5%).
- Because defects are created in the film bulk, you must analyze in details of bulk properties – FTIR and EPR data. However, FTIR spectra did not analyzed at all. I suggest to compare relative intensity (ratio) of FTIR peaks corresponding to polar and non-polar bonds as a function of fluence.
- EPR data are confusing. Why EPR peak is shifted for 3-MeV protons? I suggest to calculate the concentration of free radical for each irradiation conditions, and to compare.
- The statement on line 138-141 about large molecular weight after carbon irradiation is inconsistent with your statement on line 268-269, and contradicts to the fact that irradiation results in bond breakage, cut of the polymer chains and reduction of molecular weight of the fragments. Therefore, the decrease of the dielectric constant at high frequencies in Fig.1d has another origin.
- Additional mistakes. Line 117, It is speculation about non-polar groups, it does not follow the data. Remove it. Line 129, why voids are produced by protons whose are going through thin PI films (50 um)? It needs reference at least.
Line 139-141, It is wrong statement about “three-dimensional network”? Does non-irradiated PI structure be one-dimensional structure? The term explains nothing. Remove it.
Line 154-155, orientation polarization and dipole polarization needs an illustration such as in Fig.8.
Line 179-182, unclear English, in particular, what is “heat carriers”?
Line 249-252, did you use 3 different names for 2 free radicals?
Line 334-335, it is confusing, why free radicals of pyrolytic carbon make a compound in air? How is it possible with radicals in the PI bulk? It contradicts with your EPR data too.
Fig.8 is misleading, and must be changed.
English correction is strongly recommended.
Author Response
Dear Editor, Thank you very much for your reply and help. Thanks a lot for the reviewers’ comments and their kind suggestions of our manuscript (Polymers-717393) entitled “Low Dielectric constant Polyimide obtained by Four kinds of Irradiation sources”. We have revised the article as the reviewers’ suggestions. In order to make the changes easily viewable for you and reviewers, in the revised manuscript, we marked the revisions with red color and used “Track Changes”function in the MS as you suggest. We have checked carefully every sentences of the whole manuscript and corrected the grammar mistakes. We hope the revised paper would satisfy you and the reviewers. We are looking forward to hearing from you soon. Kind Regards, Xingji li .......................................................................................................................................... Revision list according to the comments Reviewer :1.The authors have changed the manuscript. They added the experimental details, rearranged the figures, added new results by EPR, and made English corrections in some parts. However, the presentation remains ambiguous. The main conclusion – the decrease in the dielectric constant after particle irradiations is due to reduction of amount of polar bonds – is not supported by presented results. The description of the mechanism is vague, and analysis of experimental data is misleading and suffers with many inconsistent statements. Also, flaws in English make it difficult to understand the results. The manuscript does not give meaningful information in the present form, and needs significant modification. Reply: Thank you very much for your time to second review the manuscript, and put forward a lot of good advice for it. We will modify the problems and mistakes as your comments and we found. This article mainly want to report that the low dielectric constant of PI film obtained by the four kinds of irradiation sources, this results are different with gamma irradiation which can get high dielectric constant of polymers. Last time we have explained that the results are no problem, they can be get this results by irradiation. We clear about the dielectric mechanism of the high dielectric constant, for the dielectric mechanism of low dielectric constant we provide several possible reasons, the exactly dielectric mechanism is not clear right now, and it will be our next research direction. Thank you for your understanding. We have try our best to modify this paper according to your opinion, and we believe that the second revised manuscript will be improve much better. Wish this revised manuscript will satisfy you. Thank you again for your good suggests to let the paper more better. Particular comments 1.The major critique is that there is no physically meaningful analysis of data. The evidence of specific polar bond cleavage is not given. XPS data are probing the surface composition only, and it may differ from that in the film bulk. FTIR data are valuable for analysis, but there is no detail comparison at all. EPR data are puzzling as the peak shifts for 3MeV protons. It causes doubt in correctness of the EPR measurements. Moreover, amount of free radicals are the largest after carbon irradiation as expected, but the dielectric constant change is the smallest, and the implanted carbon atoms stay in the PI film. Therefore, the cleavage of bonds is unlikely be the main effect here. You should consider alternative explanations. Also, as I said early, analysis of the mechanism has to be discussed separately for each particle. In particular, large effect of 1MeV electron irradiation need to be discussed in details. Reply: Thank you very much for your comments. This paper mainly report that the low dielectric constant of PI film obtained by the four kinds of irradiatio sources. As we all known that the XPS data resluts may have some mistake due to XPS Peak splitting translation. FTIR data, we have calculated as your requirements, but the ratio of polar group and nonpolar groups basic did not change, so we don't have added to the article, but the idea is very good, we will be use for future research. Thank you very much for your understanding. In addition, the FTIR data we put here just want show that irradiation did not produce a new group or disappear the existing group. Thank you for notice the problem, yes it is due to the EPR equipment. We have deleted the EPR data about the 3MeV protons and added the calculated the concentration of free radical for each irradiation conditions, see the reply 4. As your suggests,we have revised the explanations about the dielectric constant. For these four radiation sources, the polymer is basically degradation or cross-linking, and this experiment data shown the PI after irradiation probably degradation. From these experimental results, it is difficult to distinguish the difference between these irradiation sources for the dielectric mechanism. It will be our next research direction. In order to make the article better, let the reader understand easily, also in order to let you satisfy, we proposed several possible reasons about the dielectric mechanism. Which shown in the manuscript dielectric constant section and discussion section. We copy some here par your check. According to the evolution characteristics of the dielectric properties of polymers with frequency, the dielectric constant of polymer are closely related to it’s characteristics of molecule, chain group structure and the internal defects[23-24]. When the frequency is lower than 1×108 Hz, the dielectric behavior of PI films depends mainly on the orientation polarization, the interface polarization and dipole polarization, as shown in figure 9. The interface polarization mainly comes from the defects after irradiation, as shown in figure 8. The orientation polarization mainly comes from the main chain of molecule. This experiment test at room temperature, so the polarization is weak. The dipole polarization mainly due to the polarization of polar groups such as C-O, C=O and C-N. The results are different from the high dielectric constant induced by irradiation such as gamma, so here may be some new polarization mechanisms leading to the reduction of dielectric constant. The sample schematic of orientation polarization, interface polarization and dipole polarization may happen in PI films are shown in figure 9. In the dielectric constant section, refer to these polarizations. The orientation polarization is mainly the molecular main chain polarization and the ability of the polarization relaxation related to the movement of main chain molecules. The molecular chains of polyimide are linear molecules is strong at high temperature and weak at room temperature. This dielectric properties results all tested at room temperature, thus the orientation polarization is weak. The interface polarization happen due to the defects produced by irradiation. 2.The studied PI films (50 um) are transparent for high-energy electrons and protons, and therefore, they should break chemical bonds and create defects in the bulk. It contradicts the irradiation process in Fig.8 where the top layer is modified. Moreover, “oxide layer” in Fig.8 is inconsistent with your statement on page 10, line 250, and the XPS data did not show surface oxidation (only change of C-O by 0.5%). Reply: Thank you very much for your comments. Considering the change of C-O content is low, we agree the the “oxide layer” is inaccurate, but the free radicals decrease as the EPR data shown, most of the free radicals on surface combine with oxygen, we modify the “oxide layer” to free radicals on the surface combine with oxygen. 3.Because defects are created in the film bulk, you must analyze in details of bulk properties – FTIR and EPR data. However, FTIR spectra did not analyzed at all. I suggest to compare relative intensity (ratio) of FTIR peaks corresponding to polar and non-polar bonds as a function of fluence. Reply: Thank you very much for your comments. We have calculated the relative intensity (ratio) of FTIR peaks corresponding to polar and non-polar bonds as a function of fluence as your opinion, but the relative intensity did not change due to the error. Thus we did not put on the manuscript, and the FTIR data we put here just want show that irradiation did not produce a new group or disappear the existing group. As the question 1 we answer you. 4.EPR data are confusing. Why EPR peak is shifted for 3-MeV protons? I suggest to calculate the concentration of free radical for each irradiation conditions, and to compare. Reply: Thank you very much for your comments. The EPR peak of 3 MeV protons do have problem due to the test equipment. We have deleted it and thank you for notice the problem. Moreover, the concentration of free radical for each irradiation conditions were calculated to compare and shown in figure 6. We copy here for your check. The figure (d) shown that the content of free radicals changes with fluence or time after irradiation. The data were calculated by the Formula 2 (2) Where N is the content of free radicals, S is the measured EPR spectrum, m is the quality of test sample. As can be seen from the figure(d), the PI films produce a lot of free radicals and increases with the irradiation fluences after the electron and carbon ions irradiation, moreover, more free radicals produce after carbon ion irradiation than after electron irradiation. (a) (b) (c) (d) Figure 6. EPR curves of pristine and irradiated PI films with different source and fluences (a)1MeV electron, (b) annealing after 1MeV electron, (c)25MeV carbon ion, (d) change of content of free radicals. 5.The statement on line 138-141 about large molecular weight after carbon irradiation is inconsistent with your statement on line 268-269, and contradicts to the fact that irradiation results in bond breakage, cut of the polymer chains and reduction of molecular weight of the fragments. Therefore, the decrease of the dielectric constant at high frequencies in Fig.1d has another origin. Reply: Thank you very much for your comments. We have deleted the line 138-141 and given some possible reasons for dielectric constant list in dielectric constant section, as the question 1 we answer you. 6.Additional mistakes. Line 117, It is speculation about non-polar groups, it does not follow the data. Remove it. Line 129, why voids are produced by protons whose are going through thin PI films (50 um)? It needs reference at least. Reply: Thank you very much for your comments.We have deleted the line 117 and line 129. Line 139-141, It is wrong statement about “three-dimensional network”? Does non-irradiated PI structure be one-dimensional structure? The term explains nothing. Remove it. Reply: Thank you very much for your comments. We have removed it. Line 154-155, orientation polarization and dipole polarization needs an illustration such as in Fig.8. Reply: Thank you very much for your comments. We have added the sample schematic. We copy here for your check. The sample schematic of orientation polarization, interface polarization and dipole polarization may happen in PI films are shown in figure 9. In the dielectric constant section refer to these polarizations. The orientation polarization is mainly the molecular main chain polarization and the ability of the polarization relaxation related to the movement of main chain molecules. The molecular chains of polyimide are linear molecules is strong at high temperature and weak at room temperature. This dielectric properties results all tested at room temperature, thus the orientation polarization is weak. The interface polarization happen due to the defects produced by irradiation. Figure 9 Simple schematic of orientation polarization, interface polarization and dipole polarization may happen in PI films. Line 179-182, unclear English, in particular, what is “heat carriers”? Reply: Thank you very much for your comments. We have changed it, shown on lin175. Line 249-252, did you use 3 different names for 2 free radicals? Reply: Thank you very much for your comments. We have changed the name, shown on line 254 and 257. Line 334-335, it is confusing, why free radicals of pyrolytic carbon make a compound in air? How is it possible with radicals in the PI bulk? It contradicts with your EPR data too. Reply: Thank you very much for your comments. In fact, we would say that the free radicals on the surface of irradiated PI films combine with oxygen from air. We rewrite it shown on line 243-246. Fig.8 is misleading, and must be changed. Reply: Thank you very much for your comments. We have modified the figure 8, shown as the answer question 2. English correction is strongly recommended. Reply: Thank you very much for your comments. We have revised the English correction. Such as line 17,line19, line 67, line 203, line 238, line 239, line 240, lin243.

Reviewer 2 Report
The manuscript has been improved considerably but English and typos should be corrected through the manuscript before acceptance. Few examples are given bellow:
- Line 68: “n” should be in subscript.
- Line 209: “radicls” should read “radicals”
- Line 239: “show” should read “shown”
- Line 242: “increase” should read “increases”.
- Line 243: “The content of free radical change a lot” should read “The content of free radicals changes a lot”
- Line 245: “its compound” ?
- Line 249: “hydrocarbyl super oxygen free radical” ???
Author Response
Dear Editor,
Thank you very much for your reply and help. Thanks a lot for the reviewers’ comments and their kind suggestions of our manuscript (Polymers-717393) entitled “Low Dielectric constant Polyimide obtained by Four kinds of Irradiation sources”. We have revised the article as the reviewers’ suggestions. In order to make the changes easily viewable for you and reviewers, in the revised manuscript, we marked the revisions with red color and used “Track Changes”function in the MS as you suggest. We have checked carefully every sentences of the whole manuscript and corrected the grammar mistakes. We hope the revised paper would satisfy you and the reviewers.
We are looking forward to hearing from you soon.
Kind Regards,
Xingji li
..........................................................................................................................................
Revision list according to the comments
The manuscript has been improved considerably but English and typos should be corrected through the manuscript before acceptance. Few examples are given bellow:
- Line 68: “n” should be in subscript.
- Line 209: “radicls” should read “radicals”
- Line 239: “show” should read “shown”
- Line 242: “increase” should read “increases”.
- Line 243: “The content of free radical change a lot” should read “The content of free radicals changes a lot”
- Line 245: “its compound” ?
Reply:
Thank you for your remind, we have modified all the mistakes you suggests and other mistake we found. See line 67, line 203, line 238, line 239, lin 240, lin243. Thank you for reviewing and accepting the manuscript.

Round 3
Reviewer 1 Report
The authors have made corrections to the manuscript appropriately. However, the manuscript is not ready for publishing. Numerous flaws in English make it difficult to understand the text in many parts. The authors must use English proofing tools of MS Word or other services to correct grammar mistakes. It can not be published without English correction.
Other corrections:
The discussion is incomplete. You must compare your results with other works such as one after gamma-ray irradiation. What is fundamental difference between particle irradiation and gamma-ray irradiation? Compare irradiation fluence, energy, exposure depth for your and gamma-ray cases in details. What was the mechanism of degradation in case of gamma-ray irradiation in that work? How does it compare with your results – differences and similarities?
Additionally, you must give your opinion on the dominant mechanism to lowering of the dielectric constant. What mechanism - cross-linking by C-C re-bonding or destruction by bond breaking degradation - takes place in your experiments?
Last sentence in Abstract -> remove.
Line 44, the Greek symbols – explain.
Line 159-161, put it in discussion.
Line 254, 259, 263 – figure number is missing.
Line 349, “new polarization mechanism” – remove.
Equation (2), you need to show units to “N”, “S”, “ds”. Give value to factor “m”.
Fig.2(d), what is the box on the graph?
Fig. 6(d), units of scales are missing.
Fig.8, What are the chemical schemes in a box? Are they pyrolytic carbon radicals? It must be stated in the caption.
Fig.9, It is confusing. Interface polarization – what are polygons in volume – voids , defects? it must be clear. Defects on the surface (not in the volume) are easy for understanding. Dipole polarization – what are large circles? You should use the PI molecule structure (polar bonds of PI ) to illustrate it.
Author Response
Dear Editor,
Thank you very much for your reply and help. Thanks a lot for the reviewers’ comments and their kind suggestions of our manuscript (Polymers-717393) entitled “Low Dielectric constant Polyimide obtained by Four kinds of Irradiation sources”. We have revised the article as the reviewers’ suggestions. In order to make the changes easily viewable for you and reviewers, in the revised manuscript, we marked the revisions with red color and used “Track Changes”function in the MS as you suggest. We have checked carefully every sentences of the whole manuscript and corrected the grammar mistakes. We hope the revised paper would satisfy you and the reviewers.
We are looking forward to hearing from you soon.
Kind Regards,
Xingji li
..........................................................................................................................................
Revision list according to the comments
The authors have made corrections to the manuscript appropriately. However, the manuscript is not ready for publishing. Numerous flaws in English make it difficult to understand the text in many parts. The authors must use English proofing tools of MS Word or other services to correct grammar mistakes. It can not be published without English correction.
Reply:
Thank you very much for your time to review the manuscript. We get help from services to correct the grammar mistakes as your suggests, the revised manuscript has been modified in many places. We revised it very carefully when we check the manuscript. We hope the revised manuscript can satisfy you this time. Thank you for accepting it in advance.
Other corrections:
The discussion is incomplete. You must compare your results with other works such as one after gamma-ray irradiation. What is fundamental difference between particle irradiation and gamma-ray irradiation? Compare irradiation fluence, energy, exposure depth for your and gamma-ray cases in details. What was the mechanism of degradation in case of gamma-ray irradiation in that work? How does it compare with your results – differences and similarities? Additionally, you must give your opinion on the dominant mechanism to lowering of the dielectric constant. What mechanism - cross-linking by C-C re-bonding or destruction by bond breaking degradation - takes place in your experiments?
Reply:
Thank you very much for your suggest. We add some reports to the introduction to the let the introduction more fit the article, as the line56-64 shown. We have added the details compared with the gamma-ray irradiation and the dominant mechanism to lowering of the dielectric constant, as the line 328-line346 shown. We copy here for your check.
At present, some researchers have reported the effects of irradiation on the dielectric properties of polymers. S. Raghu et al[21]. used electron beam and gamma ray irradiated polymer electrolyte films, the resutls showed that the dielectric constant and conductivity increased with the increase of irradaition fluences. Qureshi et al[22]. found that the dielectric constant and dielectric loss of polyimide increased with the increase of irradiation flences after irradiation with 80 MeV O6+ ion. Quamara et al[23].pointed out that the dielectric behavior of polyimide was a non-monotonic evolution law with the increase of irradiation fluences after 50 MeV Si+ ion irradiation, and it was temperature-dependent. However, the irradiated polymers with low dielectric constant are rarely reported.
The dielectric constant of all samples decrease with the increase of the fluences and the rate of the decrease is not the same as all samples. Some researches have been reported that polymer electrolyte films have been used by electron beam and Gamma ray at 50 and 150 kGy, the results showed that the dielectric constant increased with the increase of irradiation fluences and temperature because of the presence of an appreciable number of defects and dipoles in the form of chain scission[21]. Different irradiation sources, fluences and irradiation rates have different effects on the structure, defects, and free radical damage of materials. The four different kinds of irradiation sources may have different mechanisms for low dielectric constant. The reasons why the four irradiation cause the decrease of dielectric constant are as follows: Firstly, the molecular structure change are due to the increase of the free volume of the molecular structure and decrease of the density of the material.,The large free volume reduces the number of polarizing groups per unit volume which can lead to the decrease of dielectric constant of the material. This should be the main reason for the decrease of the dielectric constant. Secondly, the irradiation maybe lead to defects of the PI films and the dielectric constant of air is 1, so the dielectric constant decreases. Thirdly, the dipole polarization of the irradiated PI films decreases. The irradiation effect will directly lead to the damage of the functional groups of the polymer. The polarity with large groups such as C=O, N-H, O-H, C-N etc. is probably detached from the molecular main chain. Therefore, the decrease of the polarization ability of the groups inside the polymer material lead to the decrease of the dielectric constant.
Last sentence in Abstract -> remove.Line 44, the Greek symbols – explain.Line 159-161 put it in discussion.Line 254, 259, 263 – figure number is missing.Line 349, “new polarization mechanism” – remove. Equation (2), you need to show units to “N”, “S”, “ds”. Give value to factor “m”.Fig.2(d), what is the box on the graph? Fig. 6(d), units of scales are missing.
Reply:
Thank you very much for your suggest. We have modified all the mistakes, which show in revised manuscript, line19, line42, line243-265, line254. The value m is too much data to put in text, so did not add. The box on the figure.2(d) we have removed it, it have no meaning, as the revised Fig.2(d) shown. The fig.6(d) is the content of free radicals, so no units.
Fig.8, What are the chemical schemes in a box? Are they pyrolytic carbon radicals? It must be stated in the caption.
Reply:
Thank you very much for your suggest. It’s the results of XPS. We have added to the caption, as line 307 shown.
Fig.9, It is confusing. Interface polarization – what are polygons in volume – voids , defects? it must be clear. Defects on the surface (not in the volume) are easy for understanding. Dipole polarization – what are large circles? You should use the PI molecule structure (polar bonds of PI ) to illustrate it.
Reply:
Thank you very much for your suggest. It’s defects on the surface, we explained in the manuscript, as the line313-314 shown. We modified the Fig.9 use the PI molecule structure to illustrate it.

Round 4
Reviewer 1 Report
The authors made significant improvement of the manuscript.
I have noticed some mistakes whose need corrections.
Abstract line 16, “induced chemical and free radicals” – the meaning is unclear. “Radiation induced chemical changes …”??
Line 44, “mess”-> mass?
Line 97-99, the sentence is unclear.
Line 240-241, “generate” -> appear
Line 253, generate -> generated
Fig. 9 is still unclear. It needs corrections. What are small and large circles around letters? Why they increase in electric field? An example of drawing of polar bonds is here (https://en.wikipedia.org/wiki/Carbon%E2%80%93fluorine_bond)
Also, why interface polarization charges are positive? Where are negative charges? See for example Figure 4.16 in “https://www.researchgate.net/publication/319417695_Structural_Optimization_of_SrMnO3_to_Study_Electro-Magnetic_Characteristics/figures?lo=1”
Author Response
Dear Editor, Thank you very much for your reply and help. Thanks a lot for the reviewers’ comments and their kind suggestions of our manuscript (Polymers-717393) entitled “Low Dielectric constant Polyimide obtained by Four kinds of Irradiation sources”. We have revised the article as the reviewers’ suggestions. In order to make the changes easily viewable for you and reviewers, in the revised manuscript, we marked the revisions with red color and used “Track Changes”function in the MS as you suggest. We have checked carefully every sentences of the whole manuscript and corrected the grammar mistakes. We hope the revised paper would satisfy you and the reviewers. We are looking forward to hearing from you soon. Kind Regards, Xingji li .......................................................................................................................................... Revision list according to the comments The authors made significant improvement of the manuscript. I have noticed some mistakes whose need corrections. Reply: Thank you very much for your approval. We have revised the mistakes whose need corrections. Thank you very much for accepting the article. Abstract line 16, “induced chemical and free radicals” – the meaning is unclear. “Radiation induced chemical changes …”?? Reply: Thank you very much for remind. We have modified it, which shown on line 17. Line 44, “mess”-> mass? Reply: Thank you very much for remind. We have modified it, which shown on line 43. Line 97-99, the sentence is unclear. Reply: Thank you very much for remind. We have modified it, which shown on line 97-98. Line 240-241, “generate” -> appear Reply: Thank you very much for remind. We have modified it, which shown on line 246. Line 253, generate -> generated Reply: Thank you very much for remind. We have modified it, which shown on line 263. Fig. 9 is still unclear. It needs corrections. What are small and large circles around letters? Why they increase in electric field? An example of drawing of polar bonds is here (https://en.wikipedia.org/wiki/Carbon%E2%80%93fluorine_bond) Also, why interface polarization charges are positive? Where are negative charges? See for example Figure 4.16 in “https://www.researchgate.net/publication/319417695_Structural_Optimization_of_SrMnO3_to_Study_Electro-Magnetic_Characteristics/figures?lo=1” Reply: Thank you very much for your suggests. We have modified it, which add the reason shown on 319-320 and add the conference [26]. Thank you very much for your time to review the article and accepting it.
